# Monitoring circulating cell-free HPV DNA in metastatic or recurrent cervical cancer: clinical significance and treatment implications

Zhuomin Yin[1], Tao Feng[1], Qing Xu[1], Wumin Dai[2], Maowei Ni[2], Juan Ni[1*†], Hanmei Lou[1*†]

[1]Department of Gynecologic Radiation Oncology, (Zhejiang Key Laboratory of Radiation Oncology), Zhejiang Cancer Hospital, Hangzhou Institute of Medicine (HIM), Chinese Academy of Sciences, Hangzhou, China; [2]Zhejiang Cancer Hospital, Hangzhou Institute of Medicine (HIM), Chinese Academy of Sciences, Hangzhou, China

## eLife Assessment

This study presents **useful** findings on the application of HPV cfDNA as a marker for monitoring treatment response and prognosis in patients with recurrent or metastatic cervical cancer. The evidence supporting the claims of the authors is **solid**, although inclusion of a larger number of patient samples would have strengthened the study. The work will be of interest to medics and biologists working on cervical cancer.

## Abstract

**Background:** Monitoring circulating HPV cell-free DNA (cfDNA) offers a minimally invasive method for surveillance in HPV-associated cancers, particularly cervical cancer. However, the role of dynamic HPV cfDNA monitoring in guiding clinical treatment decisions for recurrent or metastatic cervical cancer remains underexplored.

**Methods:** In this prospective pilot observational study, levels of HPV cfDNA in serum samples from 28 patients with recurrent or metastatic HPV-positive cervical cancer were measured via digital droplet polymerase chain reaction. Results for HPV cfDNA levels were matched to clinical outcomes and to serum levels of squamous cell carcinoma antigen (SCC-Ag) to assess the clinical potential of HPV cfDNA as a tumor marker.

**Results:** HPV cfDNA was detected in all 28 patients. Notably, median baseline HPV cfDNA levels varied according to the metastatic pattern observed in individual patients (p=0.019). All participants exhibited changes in HPV cfDNA levels over a median monitoring period of 2 months (range 0.3–16.9 months) prior to evaluations for treatment response or disease progression. Among 26 patients initially diagnosed with squamous cell cervical cancer, the positivity rate was 100% for HPV cfDNA and 69.2% for SCC-Ag (p=0.004, 95% confidence interval (CI), 0–0.391). Among 20 patients longitudinally monitored for squamous cell cervical cancer, the concordance with changes in disease status was 90% for HPV cfDNA and 50% for SCC-Ag (p=0.014, 95% CI, 0.022–0.621).

**Conclusions:** Our study demonstrates that HPV cfDNA is a promising tumor marker for monitoring of recurrent or metastatic HPV-positive cervical cancer.

*For correspondence:
juejue3149@163.com (JN);
louhm@zjcc.org.cn (HL)

†These authors contributed equally to this work

Competing interest: The authors declare that no competing interests exist.

**Funding:** This work was supported by the Key R&D Program of Zhejiang (2022C04001), the Zhejiang Province Medicine and Health Science and Technology Program (2020KY454), the Zhejiang Science and Technology Department Public Welfare Project (LGF22H160075).

## Introduction

Cervical cancer (CC) is the fourth most prevalent malignancy in terms of both incidence and mortality among females globally, and it is the primary human papillomavirus (HPV)-associated cancer (*Sung et al., 2021*; *Szymonowicz and Chen, 2020*). Despite advances in CC treatment, challenges persist regarding recurrence and metastasis (*Cibula et al., 2018*). The treatment landscape for recurrent or metastatic CC has changed with the emergence of targeted immunotherapy drugs. However, the lack of effective biomarkers hinders the assessment of treatment efficacy and the ability to predict patient outcomes in this setting (*Gennigens et al., 2022*). There is a pressing need for more effective and minimally invasive biomarkers for serial monitoring of treatment responses and prognosis of patient outcomes.

As a liquid biopsy modality, measurement of HPV cfDNA released from tumor cells into the bloodstream has extensive utility in optimizing various facets of cancer management, including early diagnosis (*Chan et al., 2017*), noninvasive genotyping, pretreatment assessment, drug target identification, resistance detection (*Rothwell et al., 2019*; *Zill et al., 2018*), treatment efficacy monitoring, post-treatment follow-up, and relapse prediction (*Alix-Panabières and Pantel, 2016*; *Dawson et al., 2013*; *Tie et al., 2016*). HPV is the cause of most CC cases (*Okunade, 2020*). Viral DNA from high-risk HPV genotypes integrates into host cell genomes, resulting in widespread expression of virus-specific E6/E7 proteins (*Narisawa-Saito and Kiyono, 2007*). As HPV-associated cancers release HPV cfDNA into the host's bloodstream, circulating HPV cfDNA in free or integrated form is an attractive potential biomarker that can be detected in blood (*Jeannot et al., 2016*; *Carow et al., 2017*). Typically, circulating HPV cfDNA is detected using digital droplet polymerase chain reaction (ddPCR) or next-generation sequencing (NGS) technology (*Naegele et al., 2024*). ddPCR allows direct, independent, and absolute quantification of HPV cfDNA in samples, with a DNA detection threshold as low as 1 copy/mL. Results can be obtained within 1 day; therefore, ddPCR is relatively cost-effective (*Mazurek and Rutkowski, 2023*; *Chatfield-Reed et al., 2021*).

Recent advances in targeted therapies and immunotherapies have significantly enhanced the treatment efficacy for recurrent or metastatic CC and have increased overall survival (OS) (*Gennigens et al., 2022*). Some patients may require maintenance therapy for up to 2 years, and others may survive with their tumor for an extended period. Consequently, it is imperative to closely monitor each patient's condition and adjust their treatment plans accordingly. However, conventional imaging methods occasionally fail to reflect disease changes in a timely manner (*Nabet et al., 2020*), while blood biomarkers such as squamous cell carcinoma antigen (SCC-Ag) have limited clinical effectiveness for monitoring (*Arip et al., 2022*). Considering these clinical challenges, we hypothesized that circulating HPV cfDNA levels correlate with metastatic patterns and treatment response in CC. To validate this hypothesis, we prospectively recruited an observational pilot cohort comprising patients with primary stage IVB or recurrent HPV-positive CC and measured HPV cfDNA levels. The primary objective was to examine the correlation between HPV cfDNA copy numbers, disease parameters, and treatment responses. The secondary objective was to compare the predictive value of HPV cfDNA and SCC-Ag in assessing treatment responses.

## Methods
### Study design

From August 2017 to February 2023, a total of 33 patients with pathologically confirmed primary stage IVB or recurrent HPV-positive CC were enrolled at Zhejiang Cancer Hospital. Five patients were excluded from the analysis, as outlined in *Figure 1*. The final analysis cohort comprised 28 cases: 21 with primary stage IVB CC and seven with recurrent CC. Notably, 19 cases in the primary CC group participated in the prospective clinical study (NCT03175848) initiated by us, focused on stage IVB cervical cancer. The main eligibility criteria were: (1) Pathologically confirmed diagnosis of CC; (2) Pathological evidence of at least one recurrence or metastatic lesion; (3) PCR positivity for a high-risk

**eLife digest** Cervical cancer is a serious threat to women's health, mainly caused by HPV. While treatments have advanced, monitoring treatment success remains difficult, particularly for cancer that returns or spreads. A promising method is blood testing for HPV DNA fragments from tumor cells, offering a potentially better way to track disease status.

This type of test, known as a "liquid biopsy," is less invasive and can give quicker results than traditional imaging methods. It uses a specific technology, called digital droplet PCR, which can detect even small amounts of HPV DNA in the blood. By tracking changes in HPV DNA or cell-free DNA (cfDNA), doctors can more effectively monitor cancer progression, treatment effectiveness, and potential resistance to treatment, thereby providing more personalized care for patients.

Yin et al. set out to determine if HPV cfDNA levels could predict treatment success and the progression of cancer in patients with metastatic or recurrent HPV-positive cervical cancer. The researchers also compared HPV cfDNA to another blood marker, the squamous cell carcinoma antigen (SCC-Ag), to determine which is a better predictor of treatment response.

The results showed that HPV cfDNA can help predict how well treatment is working, how the disease progresses, and the likelihood of recurrence. HPV cfDNA proved to be more accurate than SCC-Ag for monitoring treatment. This suggests that HPV cfDNA could be a useful marker for tracking cancer and planning long-term care, but more research is needed to confirm these findings.

The study of Yin et al. could help patients with advanced HPV-related cancers. As new treatments and drugs continue to emerge, dynamic HPV monitoring could guide doctors in choosing the most effective treatment for these patients. However, more large-scale studies are needed to confirm these findings before they can be widely used.

---

HPV subtype in pretreatment exfoliated cervical cells or a serum sample. All patients consented to the study protocol, including collection of blood samples throughout the study. Patients received chemotherapy with or without immunotherapy, targeted therapy, or radiotherapy (RT). Exclusion criteria: (1) Non-HPV-related cervical cancer; (2) No measurable lesions; (3) Synchronous or multiple cancers; (4) Patient refusal to provide blood samples; (5) Patient refusal to undergo regular follow-up and periodic imaging evaluations. For patients with primary stage IVB CC who had a single sample collected, the sample was obtained at treatment initiation. For the group undergoing longitudinal sampling, patients with primary stage IVB CC had three to five blood samples collected at treatment initiation, mid-treatment, and during follow-up, while patients with recurrent CC had three blood samples collected during treatment, starting from enrollment. For all serum specimens, HPV cfDNA was quantified using ddPCR. For patients with squamous cell CC in the sequential sampling group, concurrent SCC-Ag testing was performed at a time point that matched, or was within 7 days before or after, the HPV cfDNA sampling.

The 2018 International Federation of Gynecology and Obstetrics (FIGO) staging criteria were applied. Each patient underwent routine imaging assessments, and treatment efficacy was evaluated according to Response Evaluation Criteria in Solid Tumors (RECIST version 1.1). During treatment, imaging evaluations are typically conducted every two cycles of chemotherapy. Imaging assessments are also conducted before and after radiotherapy. For patients receiving immunotherapy or targeted therapy maintenance, assessments should be performed at least every 3 months. In follow-up, imaging is recommended every 3 months for the first 2 years and every 6 months from years 3–5, though this schedule may be adjusted based on individual patient needs. We defined lymph node metastasis as metastasis in para-aortic lymph nodes or distant lymph nodes, such as supraclavicular, inguinal, or mediastinal nodes. We classified patterns of CC recurrence or metastasis into five groups: local recurrence (LR); lymph node metastasis (LNM); hematogenous metastasis (HM); lymph node + hematogenous metastasis (LN + HM); and lymph node + hematogenous + diffuse serosal metastasis (LN + H + DSM). DSM encompasses metastases to the peritoneal, pleural, or pericardial regions. Prior to initiating treatment with immune checkpoint inhibitors (ICIs), tumor tissue specimens should be tested for PD-L1 expression using immunohistochemistry (IHC), provided that tumor samples are available. The PD-L1 IHC 22C3 pharmDx assay is used for testing, and PD-L1 expression in cervical cancer is quantified by the Combined Positive Score (CPS). The Ethics Committee of Zhejiang Cancer Hospital approved the study.

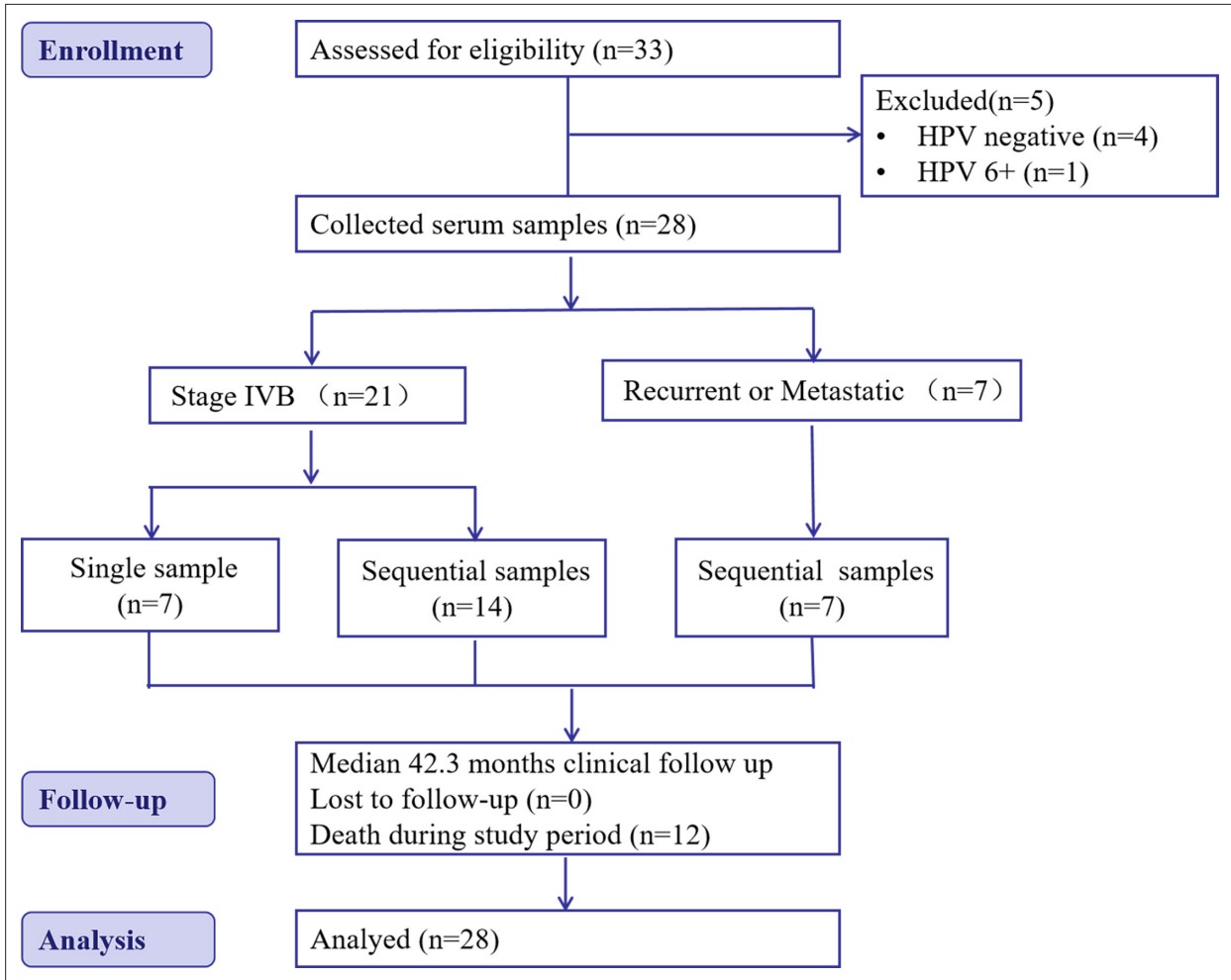

**Figure 1.** CONSORT flow diagram showing the enrollment of an observational cohort of patients diagnosed with recurrent or metastatic human papillomavirus (HPV)-positive cervical cancer undergoing serum HPV cell-free DNA surveillance.

## HPV diagnostics and typing

HPV genotypes were determined via routine PCR for exfoliated cervical cells or serum samples collected before treatment. Patients were excluded from the study if both samples tested negative. A commercial PCR testing kit (Hybribio HPV Genotyping Detection Kit), designed primarily for qualitative genotyping of HPV DNA extracted from exfoliated cervical cells, was routinely used. HPV genotyping detection was performed using PCR combined with the hybridization method. This kit can detect 21 HPV genotypes, including eight low-risk genotypes and 13 high-risk genotypes. The 13 high-risk genotypes include: HPV16, HPV18, HPV31, HPV33, HPV35, HPV39, HPV45, HPV51, HPV52, HPV56, HPV58, HPV59, and HPV68.

The following steps outline the procedure: A disposable cervical exfoliation brush is placed at the cervical opening and rotated gently clockwise five times. The brush is then transferred to a sample tube containing 10 mL of cell preservation solution. The tube is stored at 4°C and tested within 1 week. The sample is centrifuged at 3000 rpm for 15 min, and the supernatant is discarded. Next, 200 µL of cell preservation solution is added to resuspend the cells. DNA is extracted using the QIAamp DNA Mini Kit (Qiagen, Hilden, Germany), and PCR amplification is performed according to the manufacturer's instructions (total reaction volume: 20 µL/sample), followed by setting baseline and threshold values for result interpretation. One patient tested positive for HPV type 6 via PCR on cervical exfoliated cells, with no high-risk HPV genotypes detected. IHC for P16 on CC tissue was negative, leading to the classification of the tumor as non-HPV-associated and exclusion from the study.

## Samples

Serum was extracted from peripheral whole blood samples at the Radiobiology Laboratory of Zhejiang Cancer Hospital. A 5 mL aliquot of whole blood was collected into a yellow-top blood collection tube and allowed to clot at room temperature for 30 min. After centrifugation at 2000 × g for 10 min in a refrigerated centrifuge, the serum was carefully transferred into polypropylene tubes in 1 mL aliquots and stored at −80°C. Frozen serum samples were subsequently transported to the Oncology Research Institute of Zhejiang Cancer Hospital for DNA extraction.

## Procedure for ddPCR analysis

Before analysis, serum samples were thawed and centrifuged at 2000 × g at 4°C for 10 min for DNA extraction. In accordance with the manufacturer's protocol, HPV cfDNA was isolated from 2 mL of serum using a QIAamp Circulating Nucleic Acid Extraction Kit (Qiagen, Hilden, Germany). The DNA was eluted twice through a column for purification, resulting in 60 μL of eluate that was stored at −80°C until analysis. Primers and probes for ddPCR detection were designed based on E7 gene sequences of the target HPV genotypes to generate amplicons of varying length (*Supplementary file 1*). Each ddPCR reaction used 30 μL of DNA template. According to the manufacturer's instructions for the QIAcuity QX-200 ddPCR platform (Qiagen), ddPCR reactions consisted of 40 μL of reaction mixture per well that included the primers, probes, and template. The reactions were amplified in QIAcuity 26,000 24-well Nanoplates (Qiagen) under the following conditions: initial enzyme activation at 95°C for 2 min, followed by 50 cycles of denaturation at 95°C for 15 s and annealing at 60°C for 30 s. The exposure time for imaging of partitions was 400 ms for fluorescein amidite (FAM) and 300 ms for 2'-chloro-7'-phenyl-1,4-dichloro-6-carboxyfluorescein (VIC). Data analysis was performed using QIAcuity software version 2.1.7 (Qiagen) to quantify HPV copy numbers.

## HPV cfDNA monitoring protocol

Participants were enrolled on a rolling basis, and the first serum sample collected was regarded as the baseline sample. A baseline sample was categorized as a treatment initiation sample if it was collected between Day−14 and Day+30 preceding initial treatment in patients with primary stage IVB CC, or before treatment for relapse or disease progression in patients with recurrent CC.

For patients with sequential samples, the time at which the initial blood sample was collected was designated as time 0. Up to four additional blood samples were collected at intervals ranging from 8 to 962 days, with a median of 73 days and a mean of 128 days, and blood collection continued for up to 1513 days. Blood sample collection was coordinated with patient treatment and follow-up times to the greatest extent possible for measurement of HPV cfDNA levels (copies/mL). Serum SCC-Ag levels are routinely assessed multiple times before, during, and after treatment for patients with squamous cell CC at our hospital using an Abbott Architect instrument (Abbott Laboratories, Abbott Park, IL, USA). To ensure consistency in the analysis, we selected the SCC-Ag values measured simultaneously with the HPV serum samples, or chose the most recent available SCC-Ag values within 7 days. SCC-Ag levels < 1.5 ng/mL were classified as normal, while levels >= 1.5 ng/mL were classified as elevated. The upper limit of detection was 70 ng/mL. For statistical purposes, clinical test results exceeding 70 ng/mL were treated as 70 ng/mL.

Following the literature (*Jeannot et al., 2021*; *Cabel et al., 2018*), serum samples were deemed HPV-positive if at least three droplets containing HPV amplicons were identified. Samples with fewer than three droplets containing HPV amplicons or no amplicons detected were categorized as HPV-negative. Serum HPV cfDNA levels were quantified as copies/mL.

## Statistical analyses

Statistical analyses were performed using GraphPad Prism 9. The Mann-Whitney U test was used to assess the difference in the number of viral DNA copies between two groups or two metastasis patterns. Kendall's $\tau$ correlation test was used to determine the coefficient of correlation between two factors. Rate comparisons were conducted using Fisher's exact test. Kaplan-Meier survival analysis was performed to calculate the hazard ratio with a 95% confidence interval (CI), using the Cox model. Overall survival (OS) was defined as the time from diagnosis until death from any cause or the last follow-up date (December 31, 2023). All p-values reported are two-tailed, with statistical significance defined as $p < 0.05$.

**Table 1.** Characteristics of patients with metastatic or recurrent cervical cancer.

| N | All patients (n=28) | Primary IVB stage (n=21) | Recurrence or metastasis (n=7) |
|---|---|---|---|
| Age (years) | | | |
| Median age (range) | 52(34-67) | 51(34-66) | 53(37-67) |
| Pathological types (n) | | | |
| Squamous cell carcinoma | 26 | 20 | 6 |
| Adenocarcinoma | 1 | 0 | 1 |
| Large cell neuroendocrine carcinoma | 1 | 1 | 0 |
| HPV subtype | | | |
| 16 | 20 | 14 | 6 |
| 58 | 3 | 3 | 0 |
| 18 | 2 | 2 | 0 |
| 31 | 1 | 0 | 1 |
| 66 | 1 | 1 | 0 |
| 16,33 | 1 | 1 | 0 |
| Baseline serum sampling time | | | |
| Treatment initiation | 25 | 20 | 5 |
| During treatment | 3 | 1 | 2 |
| Pattern of metastasis | | | |
| Local recurrence | 1 | 0 | 1 |
| Lymphatic node metastasis | 11 | 10 | 1 |
| Hematogenous metastasis | 4 | 2 | 2 |
| Lymph node +hematogenous metastasis | 9 | 8 | 1 |
| Lymph node +hematogenous + diffuse serosal metastasis | 3 | 1 | 2 |
| Treatment modality | | | |
| Neoadjuvant chemotherapy | 18 | 18 | 0 |
| Surgeries | 2 | 0 | 2 |
| Adjuvant chemotherapy | 25 | 18 | 7 |
| Radiotherapy/concurrent chemoradiotherapy | 25 | 18 | 7 |
| Targeted therapy | 12 | 7 | 5 |
| Immunotherapy | 16 | 9 | 7 |

## Results

### Patient characteristics

Our study included 28 patients diagnosed with HPV-positive advanced CC treated at our hospital, comprising 21 cases (75%) with primary stage IVB CC and seven cases (25%) with recurrence and metastasis after treatment. The clinical characteristics of the patients are listed in *Table 1*. The median age at diagnosis was 52 years (range 34–67 years). In total, 76 serum samples were obtained from the cohort, consisting of 69 longitudinal samples from 21 patients (3–5 samples per patient) and single-time samples from seven patients at treatment initiation. Of the baseline samples, 25 (89%) were obtained at treatment initiation. HPV cfDNA was quantified in all serum samples via ddPCR. Sixteen patients received ICIs, of whom 12 also received concurrent targeted therapy, while four received ICIs alone. Three patients with stage IVB disease were treated with Paclitaxel/Cisplatin (TP) + Bevacizumab (Bev) + ICIs as first-line therapy, and 13 patients received ICIs after disease progression. The specific

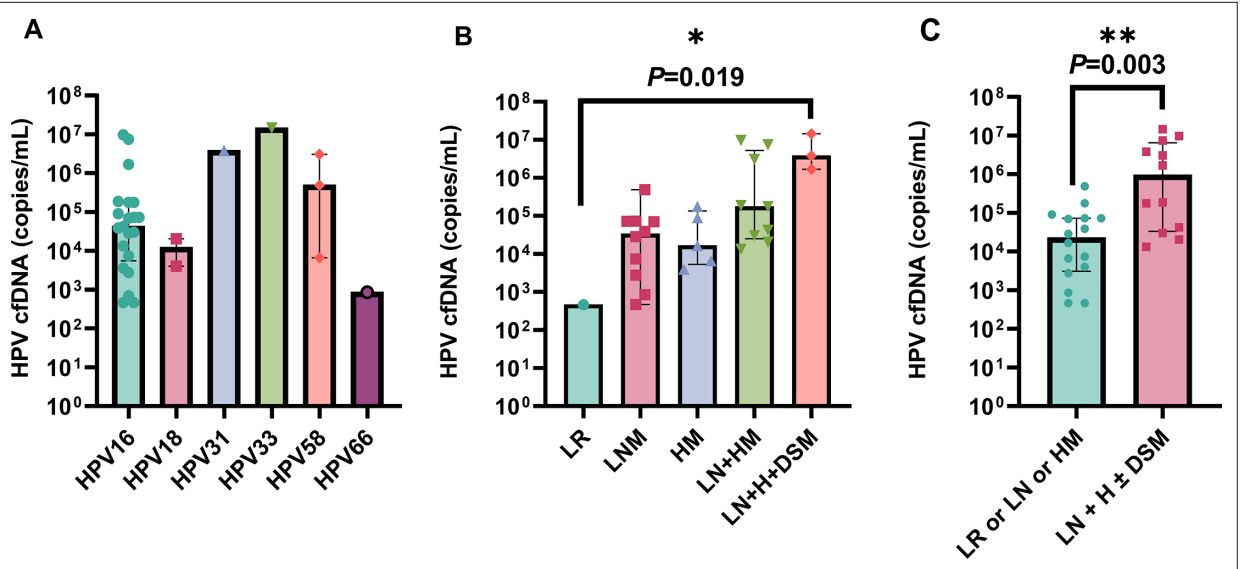

**Figure 2.** Comparison of baseline copy numbers of HPV cfDNA among different HPV genotypes and recurrence/metastasis pattern groups (n=28). (**A**) Serum CONSORT (HPV) cell-free DNA (cfDNA) copy number for six HPV genotypes at baseline. (**B**) Baseline HPV cfDNA copy numbers for five recurrence/metastasis subgroups. Statistical significance was determined using a two-sided Kruskal-Wallis test. Abbreviations: LR, local recurrence; LNM, lymph node metastasis; HM, hematogenous metastasis; LN + HM, lymph node + hematogenous metastasis; LN + H + DSM, lymph node + hematogenous + diffuse serosal metastasis. (**C**) Relationship between HPV cfDNA copy number and metastatic pattern. Statistical significance was determined using a two-sided Mann-Whitney U test. All plots show the median and interquartile range on a $\log_{10}$ scale.

treatment regimens and HPV cfDNA copy numbers for each patient are provided in *Supplementary file 2*.

## Validity of the test for HPV cfDNA detection and genotyping

To validate the consistency of HPV genotyping between serum and exfoliated cervical cells from the same patient, we conducted conventional PCR genotyping of HPV cfDNA in baseline serum. The analysis revealed that only 12/28 patients (42.9%) tested positive, however, the genotyping outcomes were entirely consistent with those obtained from matched cervical exfoliated cells. Subsequently, the 28 baseline serum samples underwent qualitative and quantitative assessment using ddPCR. The results revealed that all 28 patients (100%) tested positive for HPV cfDNA, with HPV typing showing complete concordance (100%) with PCR results from matched cervical exfoliated cells. HPV genotyping revealed that HPV16, HPV58, HPV18, and other genotypes (HPV31, HPV66, HPV33) accounted for 72.4%, 10.3%, 6.9%, and 10.3%, respectively (*Figure 2A*).

## Correlation between tumor metastasis pattern and baseline HPV cfDNA

Analysis for the study cohort revealed an association between tumor metastatic pattern and baseline HPV cfDNA levels. According to their metastatic status at baseline, the patients were categorized into two groups: the Single-Metastasis Pattern (SMP) group (LR, LNM, or HM); and the Multi-Metastatic Pattern (MMP) group (LN + H ± DSM). The baseline copy number significantly differed among the five recurrence/metastasis pattern groups (p=0.019, with Kruskal-Wallis test) and tended to increase with the degree of metastasis (*Figure 2B*). In the subsequent group comparisons, the LR group was excluded due to containing only a single value, leaving four groups for comparison. Significant differences were observed in two comparisons: LNM vs. LN +H + DSM (p=0.006) and HM vs. LN +H + DSM (p=0.036). No significant differences were found between the other groups: LNM vs. HM (p=0.768), LNM vs. LN + HM (p=0.079), HM vs. LN + HM (p=0.112), and LN + HM vs. LN +H + DSM (p=0.145), as determined by the Mann-Whitney U test (*Figure 2B*). The median baseline HPV cfDNA copy number was significantly higher in the MMP group than in the SMP group (p=0.003, with the Mann-Whitney U test, *Figure 2C*).

Baseline samples for patients with primary stage IVB cervical cancer were collected between Day –14 to + 30 relative to the initial treatment. Some patients provided pre-treatment samples, while others provided post-treatment samples, with the latter potentially influenced by the treatment itself. To assess whether there was a significant difference in baseline HPV cfDNA levels before and after treatment (within 30 days), a statistical analysis was performed. To minimize the impact of HPV genotypes, we focused on the baseline HPV cfDNA values of 14 HPV16-positive patients with primary stage IVB CC (six pre-treatment and eight post-treatment samples). The median HPV cfDNA value for pre-treatment samples was $8.01 \times 10^4$ copies/mL (range $7.5 \times 10^3$–$9.7 \times 10^6$ copies/mL), while the median for post-treatment samples was $3.5 \times 10^4$ copies/mL (range $4.7 \times 10^2$–$7.4 \times 10^6$ copies/mL). The difference between the two groups was not statistically significant (p=0.414, with the Mann-Whitney U test).

## HPV cfDNA as a predictor of treatment response or failure

We followed the enrolled patients for an average of 24.1 months (range 2.1–77.8 months). *Figure 3* shows temporal changes in serum HPV cfDNA levels. In all patients with longitudinal testing, changes in HPV cfDNA levels occurred at a median of 2 months (range 0.3–16.9 months) before imaging confirmation of a treatment response or disease progression. Six patients with squamous cell carcinoma who experienced clinical disease progression during treatment (as per RECIST) exhibited an elevation in HPV cfDNA copy numbers before imaging-confirmed progression. The median time from detection of elevated plasma HPV cfDNA to imaging confirmation of disease progression was 4.2 months (range 1.9–16.9 months; *Figure 3A1*). Similarly, we observed a consistent decrease in HPV cfDNA copy numbers in 16 patients before imaging confirmed a treatment response. The median time from detection of a decrease in HPV cfDNA to imaging confirmation of disease regression was 1.2 months (range 0.3–2.8 months; *Figure 3B1–D1*).

For patients in whom systemic cytotoxic chemotherapy was effective, a significant decrease in HPV cfDNA levels could be detected after chemotherapy. For two patients with high HPV cfDNA levels of $3.1 \times 10^6$ copies/mL and $1.7 \times 10^6$ copies/mL at baseline, the HPV cfDNA copy number decreased significantly following one or two cycles of paclitaxel + cisplatin (TP) chemotherapy (median 98.2%, range 96.7–99.5%). The rate of decline in viral load slowed after the subsequent cycle of TP chemotherapy, with a median decline of 51.3% (range 39–63.5%). In addition, a patient with baseline HPV cfDNA of $2.8 \times 10^4$ copies/mL experienced a 45% reduction in copy number following one cycle of TP chemotherapy (*Figure 3B1*).

We observed that changes in HPV cfDNA levels may indicate a response to combined immunotherapy and targeted therapy. One patient diagnosed with primary stage IVB CC and multiple metastases (LN + H + DSM) tested positive for HPV33 and HPV16 in exfoliated cervical cells and serum. The patient received four cycles of TP and concurrent chemoradiotherapy plus brachytherapy for the pelvic-abdominal primary focus, followed by two additional TP cycles. The HPV33 viral load decreased from a pretreatment level of $1.5 \times 10^7$ copies/mL to $6.7 \times 10^4$ copies/mL after treatment; however, the HPV16 viral load increased from $7.7 \times 10^2$–$1.2 \times 10^4$ copies/mL. Subsequent imaging evaluation after 58 days indicated the emergence of new foci in the lungs, prompting adjustments to the patient's treatment plan. Guided by the immunohistochemical presence of PD-L1-positive cells (CPS = 10) in the primary cervical lesion, the patient received one cycle of TP + Bev + ICIs treatment. This regimen led to a swift decrease in viral levels of both HPV genotypes after 36 days, with HPV 33 reaching undetectable levels and HPV 16 decreasing to $7.8 \times 10^2$ copies/mL (*Figure 3C1*). Three patients with stage IVB disease received TP + Bev + ICIs as first-line therapy. Two patients achieved complete remission (CR) (CPS = 70 and CPS = 5), while one experienced disease progression (CPS = 1). HPV cfDNA levels closely correlated with the clinical outcomes of these patients, demonstrating a sensitive and effective response (*Supplementary file 2*).

Serum HPV cfDNA levels also changed in response to RT. A transient rise (20.8-fold) in HPV cfDNA copy number was observed in a patient with stage IVB CC who experienced a 2-week interruption of RT due to grade IV thrombocytopenia. Following Stereotactic Body Radiotherapy (SBRT) to lung metastases, a patient with pulmonary oligometastases experienced a 29% decrease in HPV cfDNA copy number. Another patient, diagnosed with stage IVB CC and pelvic bone metastases, who underwent radical concurrent chemoradiotherapy targeting both the primary and metastatic lesions, exhibited a notable reduction in HPV cfDNA levels (*Figure 3D1*).

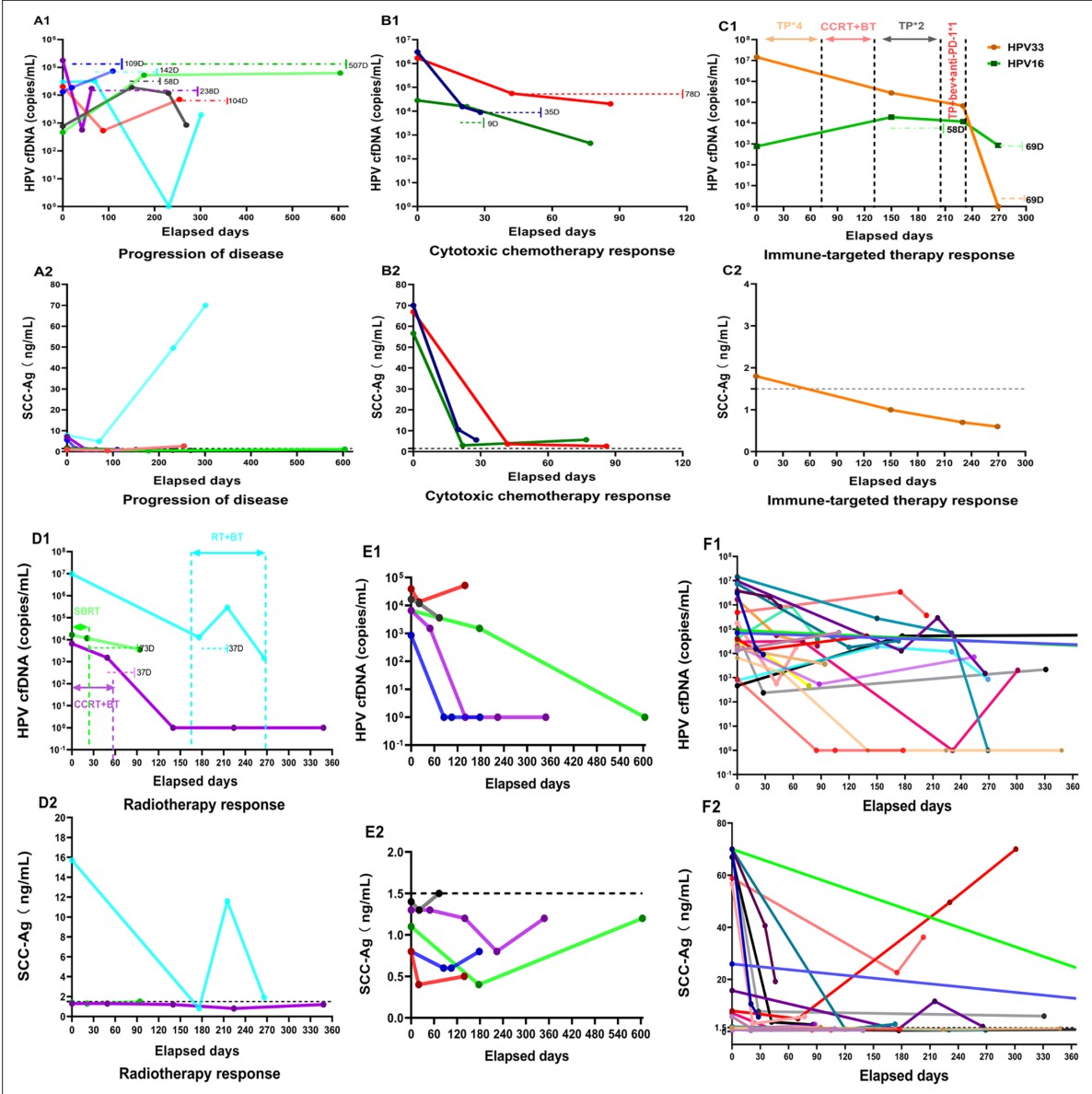

**Figure 3.** Paired plots for serum human papillomavirus (HPV) cell-free DNA (cfDNA) (copies/mL) and squamous cell carcinoma antigen (SCC-Ag) (ng/mL) levels were measured longitudinally in patients with metastatic or recurrent HPV-positive cervical cancer (each patient is represented by a line of the same color). HPV cfDNA levels in serum were scaled using log10. (**A1**, **A2**) Patients whose disease progressed during treatment. (**B1**, **B2**) Selected patients who showed a response to cytotoxic chemotherapy. (**C1**, **C2**) Response to immune therapy in a patient with stage IVB cervical cancer positive for HPV genotypes 33 and 16, treated with paclitaxel + cisplatin, concurrent chemoradiotherapy, brachytherapy, and bevacizumab. (**D1**, **D2**) Selected patients with a response to radiotherapy. (**E1**, **E2**) HPV cfDNA levels were significantly higher than normal at some of the longitudinal time points in five patients, but SCC-Ag levels were within the normal range (<1.5 ng/mL) at all time points. (**F1**, **F2**) HPV cfDNA levels for all patients with longitudinal samples (n=21) and matched SCC-Ag levels for the patients with squamous cell cancer (n=20). Each colored line corresponds to one patient, except in **C**. Horizontal dashed lines indicate the days between plasma HPV cfDNA levels suggestive of a response or progression (rise or fall in levels) and imaging-confirmed changes in disease status. The associated number of days (**D**) is listed adjacent to the horizontal dashed line for comparison. It should be noted that not all HPV cfDNA data points are plotted for each patient.

## Correlation between HPV cfDNA and SCC-Ag

There were 26 patients with squamous cell CC in the study cohort. All 26 (100%) had elevated serum HPV cfDNA at baseline, but only 18/26 patients (69.2%) had elevated SCC-Ag at baseline (p=0.004, 95% CI, 0–0.391, with Fisher's exact test). Among 72 serum samples from patients with squamous cell CC, the median HPV cfDNA level was $1.7 \times 10^4$ copies/mL (range $0–1.4 \times 10^7$ copies/mL) and the median SCC-Ag level was 2.6 ng/mL (range 0.4–70 ng/mL). There was no significant correlation

between SCC-Ag and HPV cfDNA levels ($R^2$=0.034, p=0.120, with Kendall's $\tau$ correlation test). For patients with squamous cell CC who had longitudinal monitoring (*n*=20), the concordance with disease change was 90% for HPV cfDNA and 50% for SCC-Ag (p=0.014, 95% CI, 0.022–0.621, with Fisher's exact test). Comparison of matched serum HPV cfDNA and SCC-Ag levels for patients with squamous cell CC (*Figure 3A–D*) revealed that HPV cfDNA exhibited dynamic fluctuations, while serum SCC-Ag levels in the majority of patients rapidly decreased to near or below the normal range (<1.5 ng/mL) following initiation of treatment. During the course of treatment, SCC-Ag levels remained within the normal range (<1.5 ng/mL) at all time points in five patients, but matched serum HPV cfDNA showed fluctuating changes above normal values at some time points (*Figure 3E1, E2*).

### Correlation between HPV cfDNA and survival

The 5 year OS rate for the entire cohort was 42.3%, with median OS of 52.1 months at median follow-up of 42.3 months (range 10.2–88.5 months). As of December 31, 2023, there were 12 patient deaths and 20 disease progression events. Analysis of survival by HPV genotypes revealed that the difference in OS between the HPV16 +group and the non-HPV16 +group was not statistically significant (p=0.052, with log-rank test; *Figure 4A*). Univariate and multivariate analyses were conducted to evaluate the association between clinicopathologic factors and patient OS, with no factors identified as significantly impacting OS. The results of the univariate analysis are shown in *Supplementary file 3*. Correlation analysis of baseline HPV cfDNA copy number with mortality and OS outcomes revealed no significant association ($R^2$=–0.111, p=0.486; $R^2$=–0.037, p=0.782, with Kendall's $\tau$ correlation test).

Patients with primary stage IVB CC (n=21) were stratified by baseline median HPV cfDNA level using $3.9 \times 10^4$ copies/mL as the dichotomization threshold. The difference in OS between the groups with $\geq 3.9 \times 10^4$ copies/mL and $<3.9 \times 10^4$ copies/mL at baseline was not statistically significant (p=0.111, with log-rank test; *Figure 4B*). Survival analysis of 21 consecutively monitored patients, categorized by the trend in HPV cfDNA levels, revealed two groups: 12 patients with decreasing levels and nine with increasing levels. The OS difference between these groups was not significant (p=0.866, with log-rank test). Additionally, patients were divided into two groups based on whether HPV cfDNA levels decreased to normal: three patients with normalization and 18 without. The OS difference between these groups was also not significant (p=0.590, with log-rank test).

## Discussion

We conducted a prospective pilot observational study in patients with metastatic or recurrent CC to analyze ddPCR HPV cfDNA results in relation to SCC-Ag levels, clinical treatment responses, and prognosis. The study findings confirm the significant clinical potential of dynamic HPV cfDNA surveillance for CC. First, we observed a correlation between baseline HPV cfDNA copy number and recurrence/metastasis patterns. Second, dynamically monitored HPV cfDNA levels appeared to predict treatment

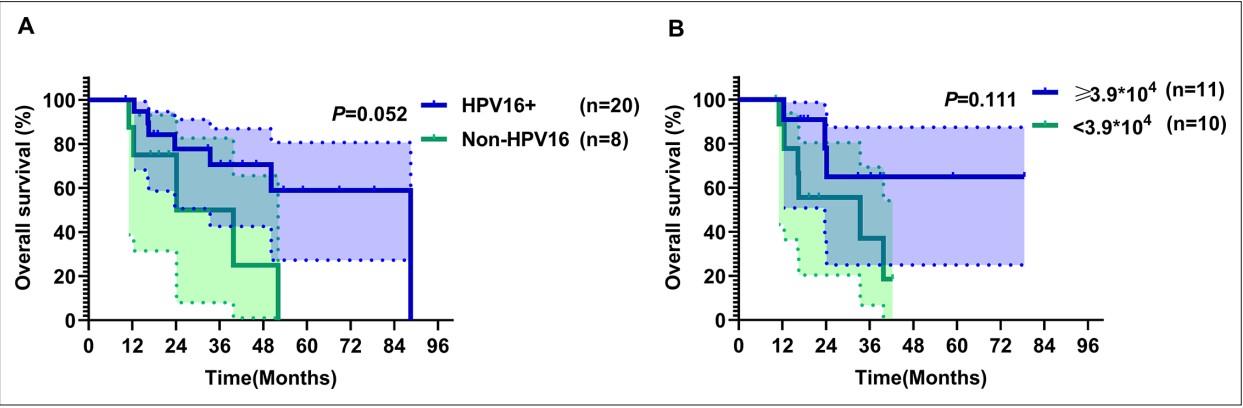

**Figure 4.** Comparison of overall survival (OS) in the entire cohort of patients (n=28) grouped by different HPV genotypes, and in stage IVB patients (n=21) grouped by different baseline serum HPV cfDNA copy numbers. (**A**) Overall survival for patients stratified by human papillomavirus (HPV) subtype (HPV 16+ versus non-HPV16+). (**B**) Overall survival for patients stratified by baseline serum HPV cell-free DNA (cfDNA) copy number, using a cutoff of 3.9 × 10⁴ copies/mL. *P*-values were calculated using a two-sided log-rank test.

response and disease progression. Finally, in monitoring HPV-associated recurrent and metastatic cervical cancer, HPV cfDNA may offer advantages over SCC-Ag.

Our positivity rate for HPV cfDNA was 42.9% (12/28) when using PCR and 100% (28/28) when using ddPCR. We found 100% agreement in HPV typing results between exfoliated cervical cells and serum samples. These findings showed the high sensitivity and specificity of HPV cfDNA detection via ddPCR. A meta-analysis comparing the accuracy of different methods for HPV cfDNA detection in HPV-positive tumors revealed that NGS outperformed ddPCR and quantitative PCR in terms of sensitivity, while specificity remained consistent across all three methods (*Naegele et al., 2024*). We found that HPV cfDNA positivity in HPV-positive CC correlated positively with tumor stage, tumor load, and lymph node status. Consistent with our findings, a recent study demonstrated a serum HPV cfDNA positivity rate of 100% in HPV-positive metastatic CC (*Kang et al., 2017*). Consequently, HPV cfDNA appears to be an ideal serum tumor marker for HPV-positive metastatic or recurrent CC, given the highly sensitive and specific detection methods available.

Studies have shown that HPV cfDNA levels correlate with disease stage, tumor size, tumor load, and lymph node status (*Cabel et al., 2021*; *Mittelstadt et al., 2023*; *Thangarajah et al., 2023*). Accurate quantification of tumor load can be challenging, particularly in settings involving diffuse serosal metastases. Therefore, we classified recurrence/metastasis patterns into five categories and observed a significant difference in median baseline HPV cfDNA copy number among these (p=0.019). Patients were categorized into SMP and MMP groups according to their recurrence/metastasis status at baseline. The MMP group had a higher median HPV cfDNA copy number at baseline than the SMP group (p=0.003). Preliminary results indicate that baseline HPV cfDNA levels may be linked to recurrence/metastasis patterns, potentially reflecting tumor burden and spread. A study by Mittelstadt et al. involving 35 patients with CC also revealed a correlation between HPV cfDNA levels and tumor load and spread (*Mittelstadt et al., 2023*). Patients with multiple metastases are likely to have a higher tumor load with greater shedding of HPV-containing DNA fragments, which can enter the blood circulation via several pathways, resulting in higher HPV cfDNA levels. However, this analysis has several limitations. It did not account for variations in cfDNA detection across HPV genotypes, which may arise from differences in detection rates or expression levels. Subgroup analysis by HPV genotype was not feasible due to the small sample size. These inherent differences may introduce bias, potentially acting as a confounding factor. Future studies will expand the sample size and include subgroup analyses by HPV genotype to better elucidate the relationship between HPV cfDNA copy number and recurrence/metastasis patterns.

In our study, changes in HPV cfDNA levels frequently preceded confirmation of disease changes on imaging scans. Several studies demonstrated that changes in HPV cfDNA copy number are associated with response to therapy for HPV-positive tumors (*Kang et al., 2017*; *Mittelstadt et al., 2023*; *Thangarajah et al., 2023*). We found that the median time from the onset of a change in HPV cfDNA copy number to imaging confirmation of a treatment response or disease progression was 2 months (range 0.3–16.9 months). Another study reported analogous findings in HPV-associated oropharyngeal cancer: changes in HPV cfDNA copy number were observed at a median of 16 days (range 12–38 days) before imaging confirmation of treatment response or disease progression in all 22 patients enrolled (*Hanna et al., 2018*). These findings suggest that HPV cfDNA can serve as a sensitive marker and a valuable clinical indicator. In addition, our results indicate that HPV cfDNA can serve as a tool for monitoring the effectiveness of RT. Among patients receiving effective RT, HPV cfDNA levels gradually declined throughout the treatment; however, interruptions in RT could lead to transient increases in HPV cfDNA levels. Moreover, our results demonstrate that the combination of immunotherapy and targeted therapy, along with cytotoxic chemotherapy, enhances tumor cell death and clearance of HPV cfDNA in vivo, surpassing the effects of cytotoxic chemotherapy alone. Our findings suggest that HPV cfDNA holds promise as a tool for evaluating the effectiveness of novel therapies, such as immunotherapy and targeted therapy and informing subsequent maintenance treatment strategies for patients.

While SCC-Ag is acknowledged as a serum tumor marker for squamous cell CC (*Chen et al., 2022*; *Salvatici et al., 2016*; *Shi et al., 2023*), its clinical sensitivity and specificity for monitoring treatment responses are limited (*Kawaguchi et al., 2013*; *Fu et al., 2019*). We compared HPV cfDNA and corresponding SCC-Ag levels in 72 serum samples and found no correlation between the two data sets ($R^2$=0.03, p=0.11). HPV cfDNA exhibited several advantages as a serum tumor marker in our

study cohort. First, the positivity rate was 100% for HPV cfDNA versus 69.2% for SCC-Ag at baseline (p=0.004, 95% CI, 0–0.391). Second, changes in HPV cfDNA copy number showed greater concordance with disease progression in the group of 20 patients with squamous cell CC with longitudinal monitoring, with a concordance rate of 90% vs 50% for SCC-Ag (p=0.014, 95% CI, 0.022–0.621). Finally, HPV cfDNA provides more comprehensive information for dynamic monitoring in comparison to SCC-Ag, as SCC-Ag levels consistently remained within the normal range in some patients. Thus, HPV cfDNA may be a more useful serum marker than SCC-Ag in patients with metastatic or recurrent CC. The lack of correlation between SCC-Ag and HPV cfDNA levels can be attributed to several factors. First, the mechanisms underlying SCC-Ag and HPV cfDNA production differ, leading to distinct treatment responses and dynamic trends. Second, the detection methods vary: HPV cfDNA is quantitatively measured by ddPCR, an exponential method, while SCC-Ag is quantified by counting, resulting in significant differences in measurement outcomes. Third, the positivity rates for HPV cfDNA and SCC-Ag differ in this study, further weakening their correlation. In clinical practice, a panel of serum tumor markers is commonly used to monitor baseline conditions prior to treatment. In cases where SCC-Ag is negative, other serum markers may be positive, aiding in the assessment of treatment efficacy and disease progression. Future research into the differing treatment responses of HPV cfDNA and SCC-Ag may help optimize the use of cervical cancer serum biomarkers and their combination.

The 5 year OS rate for the study cohort was 42.3% and the median OS time was 52.1 months. This good outcome was due to active systemic treatment (chemotherapy + immunotherapy) and local treatment (RT to primary and metastatic foci). Therefore, non-invasive dynamic monitoring using serum tumor markers is important for this population. Our data showed that correlation between OS and HPV genotypes (HPV16 + vs non-HPV16+) did not reach statistical significance (p=0.052). However, we observed a trend towards better prognosis for patients with HPV16 + than for patients with positivity for other HPV genotypes, and further expansion of the sample size may yield positive results. In the analysis of baseline copy number in relation to prognosis, the difference in OS between the groups with baseline HPV cfDNA levels $\geq 3.9 \times 10^4$ and $<3.9 \times 10^4$ copies/mL was not significant (p=0.111). Previous studies reported a positive correlation between high baseline HPV cfDNA levels and poor prognosis in oropharyngeal cancer (*Adrian et al., 2023*; *Hanna et al., 2019*; *Cao et al., 2022*).

Univariate and multivariate analyses, along with correlation analyses, were performed to evaluate factors influencing overall survival (OS) and assess the potential association between HPV cfDNA copy number and patient prognosis. However, the results were not statistically significant. Several factors may have contributed to this, including the small sample size, limited blood samples per patient, and considerable heterogeneity among patients with recurrent metastatic cervical cancer. Additionally, variables such as metastatic lesion status and treatment modalities (e.g. immunotherapy, targeted therapy, or radiotherapy) could have influenced the findings. Therefore, larger studies with extended follow-up, more frequent blood sample collection, and comprehensive analyses are needed to better elucidate the relationship between HPV cfDNA copy number and OS in cervical cancer.

The results of landmark clinical trials, including GOG240, Keynote158, and Keynote826, have integrated ICIs and targeted therapy into the clinical management of recurrent or metastatic CC, significantly improving patient survival (*Tewari et al., 2015*; *Marabelle et al., 2020*; *Colombo et al., 2021*). This shift was reflected in our enrollment process, where 16 patients received ICIs. Early-phase patients were treated with conventional chemotherapy regimens. After the Keynote-158 study, 13 patients received ICIs upon disease progression, provided that their PD-L1 CPS ≥ 1. Following the Keynote-826 study, three patients with stage IVB disease (LN +HM) were treated with TP + Bev + ICIs as first-line therapy. Two patients (CPS = 70 and CPS = 5) achieved CR, and one (CPS = 1) had remission lasting 13.5 months despite progression. These results suggest that adding ICIs to first-line therapy significantly enhances efficacy in stage IVB cervical cancer. As illustrated in *Figure 3*, dynamic changes in HPV cfDNA levels in patient C indicate a substantial reduction after one cycle of TP + Bev + ICIs compared to TP chemotherapy and radiotherapy. When other serum markers, like SCC-Ag, were uninformative, HPV cfDNA served as a valuable biomarker, suggesting the potential effectiveness of combining Bev and ICIs. These findings underscore the utility of HPV cfDNA in monitoring the efficacy of ICIs and targeted therapies in HPV-associated cancers.

Our study has several limitations, including a small sample size and a heterogeneous sequential sampling protocol, which introduced variability in baseline blood sampling timing, sample quantities, and intervals between samplings during treatment. Additionally, the heterogeneity of cfDNA levels across different HPV genotypes (due to differences in expression levels or detection efficiencies) and baseline variability of HPV cfDNA within individual patients may also contribute to potential biases in the results. We are currently conducting a prospective study of stage IVB CC. Owing to the absence of literature support for HPV cfDNA sampling protocols, we designed this initial exploratory study with a small sample size to clarify the value of HPV cfDNA monitoring and explore various sampling times and intervals. Our study results suggest that monitoring of HPV cfDNA is valuable before, during, and after treatment. Assessment of HPV cfDNA levels in every chemotherapy cycle (monthly) during treatment and every 3–6 months during follow-up may be a reasonable approach.

## Conclusions

Our prospective study suggests that HPV cfDNA, with its high sensitivity and specificity, holds promise as a biomarker for monitoring treatment response and facilitating long-term follow-up in patients with recurrent or metastatic HPV-associated cervical cancer. The baseline copy number of HPV cfDNA may be associated with metastatic patterns, thereby reflecting tumor burden and the extent of spread to some extent. As a serum tumor marker, HPV cfDNA may outperform SCC-Ag in tracking disease dynamics and enabling timely assessment of treatment responses to chemotherapy, radiation, immunotherapy, and targeted therapies. These preliminary findings highlight HPV cfDNA's potential for monitoring treatment efficacy and predicting disease progression and recurrence in HPV-associated cancers. However, further validation through large-scale prospective trials is needed.

## Acknowledgements

We thank the follow-up in the follow-up room of Zhejiang Provincial Cancer Hospital. This work was supported by the Key R&D Program of Zhejiang (2022C04001), the Zhejiang Province Medicine and Health Science and Technology Program (2020KY454), the Zhejiang Science and Technology Department Public Welfare Project (LGF22H160075).

## Additional information

### Funding

| Funder | Grant reference number | Author |
| --- | --- | --- |
| Key Research and Development Program of Zhejiang Province | 2022C04001 | Hanmei Lou |
| Zhejiang province Medicine and Health Science and Technology Program | 2020KY454 | Zhuomin Yin |
| Zhejiang Science and Technology Department Public Welfare Project | LGF22H160075 | Zhuomin Yin |

The funders had no role in study design, data collection and interpretation, or the decision to submit the work for publication.

### Author contributions

Zhuomin Yin, Conceptualization, Resources, Data curation, Formal analysis, Funding acquisition, Validation, Investigation, Visualization, Methodology, Writing - original draft, Project administration; Tao Feng, Investigation, Visualization, Methodology, Writing – review and editing; Qing Xu, Data curation, Investigation, Methodology, Writing – review and editing; Wumin Dai, Investigation, Methodology, Writing – review and editing; Maowei Ni, Formal analysis, Validation, Investigation, Methodology, Writing – review and editing; Juan Ni, Data curation, Supervision, Validation, Investigation, Writing

– review and editing; Hanmei Lou, Conceptualization, Resources, Supervision, Funding acquisition, Investigation, Writing – review and editing

### Author ORCIDs
Zhuomin Yin ⓘ https://orcid.org/0000-0002-0608-4511
Tao Feng ⓘ https://orcid.org/0000-0003-0918-9770
Qing Xu ⓘ https://orcid.org/0009-0008-5107-7071
Wumin Dai ⓘ https://orcid.org/0000-0001-9799-0490
Maowei Ni ⓘ https://orcid.org/0009-0001-2342-7328
Juan Ni ⓘ https://orcid.org/0000-0001-5473-5406
Hanmei Lou ⓘ https://orcid.org/0000-0002-3234-1673

### Ethics
Human subjects: This study was approved by the Medical Ethics Committee of Zhejiang Cancer Hospital and informed consent was obtained from all subjects. Approval identifier (approval No.) : IRB-2023-753.

Reviewer #1 (Public review): https://doi.org/10.7554/eLife.101887.3.sa1
Author response https://doi.org/10.7554/eLife.101887.3.sa2

---

## Additional files

### Supplementary files
Supplementary file 1. Primer and probe sequences for ddPCR, fragment sizes, and annealing.

Supplementary file 2. HPV cfDNA Levels and Clinical Treatment and Outcomes in the Whole Group of Patients.

Supplementary file 3. Univariate analysis of prognostic factors for overall survival (n=28).

MDAR checklist

Reporting standard 1. STROBE checklist.

### Data availability
All data generated or analyzed in this study are available in the manuscript and supplementary files. The raw data for *Figures 2–4* are in *Supplementary file 2*. *Table 1* and *Supplementary files 1–3* contain the raw data files used in the study.

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
