## [Editor Report · eLife Assessment]

This study presents **useful** findings on the application of HPV cfDNA as a marker for monitoring treatment response and prognosis in patients with recurrent or metastatic cervical cancer. The evidence supporting the claims of the authors is **solid**, although inclusion of a larger number of patient samples would have strengthened the study. The work will be of interest to medics and biologists working on cervical cancer.

---

## [Referee Report · Reviewer #1 (Public review)]

Summary:

The study by Zhuomin Yin and colleagues focuses on the relationship between cell-free HPV (cfHPV) DNA and metastatic or recurrent cervical cancer patients. It expands the application of cfHPV DNA in tracking disease progression and evaluating treatment response in cervical cancer patients. The study is overall well-designed, including appropriate analyses.

Strengths:

The findings provide valuable reference points for monitoring drug efficacy and guiding treatment strategies in patients with recurrent and metastatic cervical cancer. The concordance between HPV cfDNA fluctuations and changes in disease status suggests that cfDNA could play a crucial role in precision oncology, allowing for more timely interventions. As with similar studies, the authors used Droplet Digital PCR to measure cfDNA copy numbers, a technique that offers ultrasensitive nucleic acid detection and absolute quantification, lending credibility to the conclusions.

Weaknesses:

Despite including 28 clinical cases, only 7 involved recurrent cervical cancer, which may not be sufficient to support some of the authors' conclusions fully. Future studies on larger cohorts could solidify HPV cfDNA's role as a standard in the personalized treatment of recurrent cervical cancer patients.

Comments on revisions:

Thanks for your additional efforts and for addressing my concerns.

---

## [Author Response]

The following is the authors’ response to the original reviews

**Reviewer #1 (Public review):**
Summary:The study "Monitoring of Cell-free Human Papillomavirus DNA in Metastatic or Recurrent Cervical Cancer: Clinical Significance and Treatment Implications" by Zhuomin Yin and colleagues focuses on the relationship between cell-free HPV (cfHPV) DNA and metastatic or recurrent cervical cancer patients. It expands the application of cfHPV DNA in tracking disease progression and evaluating treatment response in cervical cancer patients. The study is overall well-designed, including appropriate analyses.Strengths:The findings provide valuable reference points for monitoring drug efficacy and guiding treatment strategies in patients with recurrent and metastatic cervical cancer. The concordance between HPV cfDNA fluctuations and changes in disease status suggests that cfDNA could play a crucial role in precision oncology, allowing for more timely interventions. As with similar studies, the authors used Droplet Digital PCR to measure cfDNA copy numbers, a technique that offers ultrasensitive nucleic acid detection and absolute quantification, lending credibility to the conclusions.Weaknesses:Despite including 28 clinical cases, only 7 involved recurrent cervical cancer, which may not be sufficient to support some of the authors' conclusions fully. Future studies on larger cohorts could solidify HPV cfDNA's role as a standard in the personalized treatment of recurrent cervical cancer patients.(1) The authors should provide source data for Figures 2, 3, and 4 as supplementary material.

We greatly appreciate your evaluation of our study and fully agree with the limitations you have pointed out. We appreciate your constructive feedback. Based on your suggestions, we have made the following additions to the article. We have realized that the information provided in Figures 2, 3, and 4 is limited. Therefore, we have presented the original data from Figures 2, 3, and 4 in tabular form in Supplementary Table 2.

(2) Description of results in Figure 2: Figure 2 would benefit from clearer annotations regarding HPV virus subtypes. For example, does the color-coding in Figure 2B imply that all samples in the LR subgroup are of type HPV16? If that is the case, is it possible that detection variations are due to differences in subtype detection efficiency rather than cfDNA levels? The authors should clarify these aspects. Annotation of Figure 2B suggests that the p-value comes from comparing the LR and LN + H + DSM groups. This should be clarified in the legend. If this p-value comes from comparing HPV cfDNA copies for the (LR, LNM, HM) and (LN + HM, LN + HM + DSM) groups, did the authors carry out post-hoc pairwise comparisons? It would be helpful to include acronyms for these groups in the legend also.

We fully agree with your point regarding the need for clearer labeling of HPV genotypes in Figures 2B and 2C. If each data point could be color-coded to represent the HPV genotype, Figures 2B and 2C would be clearer and provide more information. However, we must acknowledge that due to the limitations of our current graphing software and our graphical expertise, we were unable to fully represent each HPV genotype in the figures. To address this, we have presented the data in Supplementary Table 2. This table shows the HPV genotype for each patient, the corresponding metastasis patterns, and the baseline HPV copy numbers. We hope this will address the limitation of insufficient information in Figure 2.

The point you raised regarding whether the differences in detection results might stem from variations in subtype detection efficiency rather than cfDNA levels is a valid limitation of this study. Due to the limited sample size, we did not perform subgroup analyses based on different HPV genotypes, which may have introduced bias in the results presented in Figures 2B and 2C. In response, we have added the following clarification in the discussion section (lines 416-422) and addressed this limitation in the limitations section (lines 499-502). Based on your suggestion, we believe that it is essential to expand the sample size and perform subgroup analysis of the baseline copy numbers for each HPV genotype before treatment. We hope to achieve this goal in future studies.

Thank you for your thoughtful comments regarding the statistical analyses in the study. The p-value in Figure 2B comes from the comparison among five groups, using a two-sided Kruskal-Wallis test. Your suggestion to perform post-hoc pairwise comparisons is excellent and has made the data presentation in the article more rigorous. Following your advice, we conducted pairwise comparisons between the groups. We used the Mann-Whitney U test to compare HPV cfDNA copy numbers between two groups. Since the LR group only had one value, it could not be included in the pairwise comparisons. Significant differences were observed in two comparisons: LNM vs. LN + H + DSM (P = 0.006) and HM vs. LN + H + DSM (P = 0.036). No significant differences were found between the other groups: LNM vs. HM (P = 0.768), LNM vs. LN + HM (P = 0.079), HM vs. LN + HM (P = 0.112), and LN + HM vs. LN + H + DSM (P = 0.145), as determined by the Mann-Whitney U test (Figure 2B). (Lines 258-263).

Thank you for your thoughtful suggestion regarding the inclusion of group acronyms in the legends of Figures 2B and 2C. Including the full names corresponding to the abbreviations would indeed enhance clarity. While we attempted to add both acronyms and full names to the figure legend, the full names were too lengthy and impacted the figure's presentation. Therefore, we have provided the full names corresponding to the abbreviations in the figure caption below, to help readers easily understand the abbreviations used in the figure.

(3) Interpretation of results in Figure 2 and elsewhere: Significant differences detected in Figure 2B could imply potential associations between HPV cfDNA levels (or subtypes) and recurrence/metastasis patterns. Figure 2C shows that there is a difference in cfDNA levels between the groups compared, suggesting an association but this would not necessarily be a direct "correlation". Overall, interpretation of statistical findings would benefit from more precise language throughout the text and overstatement should be avoided.

Thank you for your insightful comments regarding the interpretation of results in Figure 2 and elsewhere. We acknowledge that there are several limitations in this study, and the interpretation of the results should be more careful and cautious. Indeed, in the results section, there were issues with inaccurate wording and exaggeration. We have made revisions in the discussion section, which are presented as follows: Preliminary results indicate that baseline HPV cfDNA levels may be linked to recurrence/metastasis patterns, potentially reflecting tumor burden and spread (Lines 411-413). Additionally, we have also made changes in the conclusion section, which are presented as follows: The baseline copy number of HPV cfDNA may be associated with metastatic patterns, thereby reflecting tumor burden and the extent of spread to some extent (Lines 511-513).

(4) The authors state that six patients showed cfDNA elevation with clinically progressive disease, yet only three are represented in Figure 3B1 under "Patients whose disease progressed during treatment." What is the expected baseline variability in cfDNA for patients? If we look at data from patients with early-stage cancer would we see similar fluctuations? And does the degree of variability vary for different HPV subtypes? Without understanding the normal fluctuations in cfDNA levels, interpreting these changes as progression indicators may be premature.

Thank you for your feedback. We appreciate your thorough review and attention to detail. Six cervical squamous cell carcinoma (SCC) patients exhibited elevated HPV cfDNA levels as their clinical condition progressed. In the previous Figures 3A1 and 3A2, we only presented data from three patients, as we initially believed that displaying the cfDNA curves from three patients would offer a clearer view, while including six patients might lead to overlap and reduce clarity. However, this may have caused confusion for readers. Based on your suggestion, we have revised Figure 3A1 to include the cfDNA curves for all six patients who with squamous cell carcinoma who experienced clinical disease progression during treatment (Figure 3A1), along with the corresponding SCC-Ag curves (Figure 3A2).

Thank you for highlighting the issue of baseline variability in HPV cfDNA. This is indeed a limitation of our study, which did not address this aspect. If baseline variability is defined as changes in HPV cfDNA levels measured at different time points before treatment in the same patient, fluctuations at different time points are inevitable and objective. Following your suggestion, we have added a discussion on baseline variability in the limitations section of the manuscript to provide readers with a more objective understanding of our study's findings (Lines 501-502).In future studies, we will incorporate baseline variability into the research design to better understand pre-treatment HPV cfDNA fluctuations and provide support for clinical decision-making.

(5) It would be helpful if where p-values are given, the test used to derive these values was also stated within parentheses e.g. (P < 0.05, permutation test with Benjamini-Hochberg procedure).

Thank you for your valuable suggestions and examples. Following your advice, we have included the statistical test methods used to obtain the p-values in parentheses wherever they appear in the results section. Additionally, we have specified the statistical test methods for the p-values below the figures in the results section.

**Reviewer #2 (Public review):**
Summary:The authors conducted a study to evaluate the potential of circulating HPV cell-free DNA (cfDNA) as a biomarker for monitoring recurrent or metastatic HPV+ cervical cancer. They analyzed serum samples from 28 patients, measuring HPV cfDNA levels via digital droplet PCR and comparing these to squamous cell carcinoma antigen (SCC-Ag) levels in 26 SCC patients, while also testing the association between HPV cfDNA levels and clinical outcomes. The main hypothesis that the authors set out to test was whether circulating HPV cfDNA levels correlated with metastatic patterns and/or treatment response in HPV+ CC.The main claims put forward by the paper are that:(1) HPV cfDNA was detected in all 28 CC patients enrolled in the study and levels of HPV cfDNA varied over a median 2-month monitoring period.(2) 'Median baseline' HPV cfDNA varied according to 'metastatic pattern' in individual patients.(3) Positivity rate for HPV cfDNA was more consistent than SCC-Ag.(4) In 20 SCC patients monitored longitudinally, concordance with changes in disease status was 90% for HPV cfDNA.This study highlights HPV cfDNA as a promising biomarker with advantages over SCC-Ag, underscoring its potential for real-time disease surveillance and individualized treatment guidance in HPV-associated cervical cancer.Strengths:This study presents valuable insights into HPV+ cervical cancer with potential translational significance for management and guiding therapeutic strategies. The focus on a non-invasive approach is particularly relevant for women's cancers, and the study exemplifies the promising role of HPV cfDNA as a biomarker that could aid personalized treatment strategies.Weaknesses:While the authors acknowledge the study's small cohort and variability in sequential sampling protocols as a limitation, several revisions should be made to ensure that (1) the findings are presented in a way that aligns more closely with the data without overstatement and (2) that the statistical support for these findings is made more clear. Specific suggestions are outlined below.(1) Line 54 in the abstract refers to 'combined multiple-metastasis pattern' but it is not clear what this refers to at this point in the text.

Thank you for your detailed feedback. You are correct that the "combined multi-metastatic pattern" was not adequately explained in the abstract, which may have caused confusion. To address this, we have clarified the definitions of the combined multi-metastatic pattern and single-metastatic pattern in lines 53-55 of the manuscript. Patients with a combined multi-metastatic pattern (lymph node + hematogenous ± diffuse serosal metastasis) exhibited a higher median baseline HPV cfDNA level compared to those with a single-metastasis pattern (local recurrence, lymph node metastasis, or hematogenous metastasis) (P = 0.003).

(2) Line 90 The reference to 'prospective clinical study (NCT03175848) in primary stage IVB CC to investigate the role of radiotherapy (RT) in combination therapy' seems not to be at all relevant at this point in the text. I would limit the description of this study to the methods.

Thank you for your thoughtful and thorough review. Your suggestions are highly relevant. Upon further reflection, we recognized that this sentence was redundant in its original placement. Following your recommendation, we have removed it from this section and moved it to the methods section (Lines 109-111). The revised statement is as follows: "Notably, 19 cases from the primary CC group participated in our prospective clinical study (NCT03175848), focused on stage IVB cervical cancer."

(3) Line 56 refers to HPV cfDNA levels (range 0.3-16.9) but what units?

Thank you for your feedback regarding the manuscript format. While you highlighted this specific issue, we have since identified several other instances of omitted units in parentheses throughout the manuscript. We acknowledge that such formatting oversights can create ambiguity for readers. Following your suggestions, we have corrected all such issues in the manuscript. We greatly appreciate your careful and thorough review.

(4) Lines 247-248 claim that higher baseline HPV cfDNA levels correlated with a more substantial post-chemotherapy decrease. This correlation should be statistically validated, and the p-value should be included.

Thank you for your insightful comments, which highlighted an issue with this sentence. Upon review, I have made the necessary revisions. Since no statistical analysis was conducted and the P-value was not provided, the original sentence was imprecise. Given the small sample size, statistical analysis is not feasible. I have revised the sentence as follows: “For patients in whom systemic cytotoxic chemotherapy was effective, a significant decrease in HPV cfDNA levels could be detected after chemotherapy” (Lines 297-298).

(5) The authors mention that baseline samples were collected "between Day -14 and Day +30 preceding initial treatment." If Day -14 indicates two weeks before treatment, then this would imply some samples were taken up to 30 days post-treatment. This notation should be clarified. To what extent might outliers or more extreme values in Figure 2 driven by variability in how baseline sampling was carried out?

Thank you for your insightful comments. Undoubtedly, this is indeed a major limitation of our study. These factors could lead to a certain degree of bias in the detection data. The primary reason is that the study was conducted during the COVID-19 pandemic, making it sometimes difficult to conduct sampling regularly. In accordance with your suggestion, I have already added this part of the content to the results section of the article (Lines 266-275). We have also included the variation in baseline sampling as a limitation in the discussion section (Lines 497-499). In future studies, we will strive to improve the study design by ensuring baseline samples are collected prior to treatment, thereby enhancing the reliability of statistical and analytical results.

(6) Would be useful to amend Figure 1 to show a subset of patients with SCC and a subset of patients who underwent longitudinal monitoring.

Thank you for your detailed suggestion. Including a subset of pathological types could indeed add more information to Figure 1. However, regarding the pathological types of the patients in this group, we have listed them in Table 1 and Supplementary Table 2. Among the 28 patients, 26 are diagnosed with squamous cell carcinoma, so 92.9% of the patients in this study have squamous cell carcinoma. To avoid making Figure 1 too complex, we decided not to include the pathological type in the figure.

(7) Line 120 "a time point matching or closely following HPV cfDNA sampling" - what is the time range for 'closely following' here? A couple of hours or days after sampling?

Thank you for your detailed feedback. Based on your suggestion, we have revised the sentence as follows:

"For patients with squamous cell CC in the sequential sampling group, concurrent SCC-Ag testing was performed at a time point that matched, or was within 7 days before or after, the HPV cfDNA sampling." (Line 123-125)

(8) Lines 178-190 and lines 179-180 seem to make exactly the same point.

Thank you very much for your careful review. Indeed, these two sentences were repetitive and conveyed the same point. I have removed the previous sentence here (lines 206-207).

(9) In Figure 4, please indicate the number of patients in each group in the legend e.g. HPV16+ (n=x number of patients).

Thank you for your feedback on the details of Figure 4 and the examples provided. We have updated Figure 4 according to your suggestions and included the number of patients in each group in the figure legend.

(10) Lines 322-3 'HPV cfDNA predicted treatment response or disease progression at an earlier time point than imaging assessments' - based on the data available and the numbers of patients, I would argue that this is too bold a claim.

Thank you very much for pointing out this issue. We fully agree with your view. We have modified this sentence as follows: "Secondly, dynamically monitored HPV cfDNA levels appeared to predict treatment response and disease progression. " (Lines 391-392).